# User-Dependent Neural Sequence Models for Continuous-Time Event Data

**Alex Boyd**[1]   **Robert Bamler**[2]   **Stephan Mandt**[1,2]   **Padhraic Smyth**[1,2]
[1]Department of Statistics   [2]Department of Computer Science
University of California, Irvine
{alexjb, rbamler, mandt}@uci.edu   smyth@ics.uci.edu

## Abstract

Continuous-time event data are common in applications such as individual behavior data, financial transactions, and medical health records. Modeling such data can be very challenging, in particular for applications with many different types of events, since it requires a model to predict the event types as well as the time of occurrence. Recurrent neural networks that parameterize time-varying intensity functions are the current state-of-the-art for predictive modeling with such data. These models typically assume that all event sequences come from the same data distribution. However, in many applications event sequences are generated by different sources, or *users*, and their characteristics can be very different. In this paper, we extend the broad class of neural marked point process models to mixtures of latent embeddings, where each mixture component models the characteristic traits of a given user. Our approach relies on augmenting these models with a latent variable that encodes user characteristics, represented by a mixture model over user behavior that is trained via amortized variational inference. We evaluate our methods on four large real-world datasets and demonstrate systematic improvements from our approach over existing work for a variety of predictive metrics such as log-likelihood, next event ranking, and source-of-sequence identification.

## 1   Introduction

Event sequences in continuous time occur across many contexts, leading to a variety of data analysis applications such as forecasting consumer purchases, fraud detection in transaction data, and prediction in clinical medicine. In such data, each event is typically associated with one of $K$ event types, also known as *marks*, and a timestamp. There has been significant amount of prior work in statistics for clustering [Du et al., 2015, Xu and Zha, 2017], factorizing [Schein et al., 2015], and generative modeling of such data, typically under the framework of marked temporal point process (MTPP) models [Daley and Vere-Jones, 2007]. We are primarily interested in the third of these objectives. The multivariate Hawkes process [Hawkes, 1971, Liniger, 2009], Poisson-network [Rajaram et al., 2005], piecewise-continuous conditional intensity model [Gunawardana et al., 2011], and proximal graphic event model [Bhattacharjya et al., 2018] are some examples of the many different MTPP models previously explored. MTPP models characterize the instantaneous rate of occurrence of each event type, the so-called *intensity function*, conditioned on the history of past events. However, strong parametric assumptions in these traditional models limit their flexibility for modeling real-world phenomena.

Recent work in machine learning has sought to address these limitations via the use of deep recurrent neural networks (RNNs). These models, such as Du et al. [2016], use expressive representations for the intensity function, use event embeddings to avoid parameter explosion, and optimize the associated log-likelihood via stochastic gradient methods. A variety of approaches have been explored to address the complex mix of discrete events and continuous time that occur in real-world event

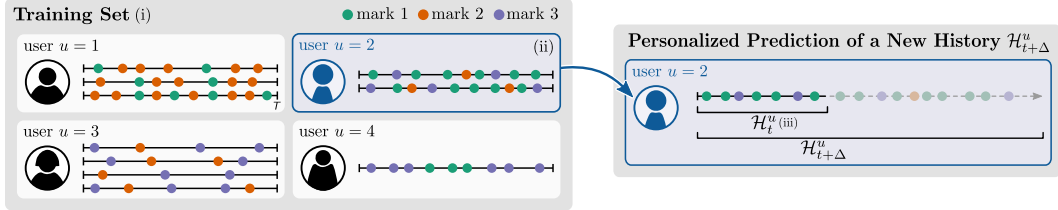

Figure 1: Personalized event sequence. We predict (or evaluate the likelihood of) the semi-transparently drawn sequence on the right conditioned on (i) a training set (left), (ii) a few reference sequences from the same user (left, blue), and (iii) a prefix $\mathcal{H}_t^u$ of the current sequence (right).

sequences [Mei and Eisner, 2017, Wang et al., 2017, Zhang et al., 2019b, Türkmen et al., 2019]. In general, these neural-based MTPPs have been found empirically to provide systematically better predictions than their traditional counterparts, due to their more flexible representations for capturing the influence of past events on future ones (both their time stamps and their types), as well as being better able to handle the significant data sparsity associated with large event vocabularies. However, a common implicit assumption in these approaches is that all of the modeled sequences originate from the same source (or user), which is often not the case in real-world datasets such as online consumer data or medical history records. Sufficiently powerful neural-based MTPPs can internally adjust for this heterogeneity after conditioning on a significant portion of a history; however, they exhibit large predictive uncertainty at the beginning of sequences. Thus, it is important to develop techniques that *personalize* predictions to account for heterogeneity across users.

To develop personalized MTPPs, we propose using variational autoencoders (VAEs) [Kingma and Welling, 2014] in conjunction with previous neural-based MTPPs. VAEs are well-suited to address the problem of personalization (e.g., Liang et al. [2018]) since they distinguish between global and local parameters, which are treated differently during training/inference. In our setup, global model parameters capture properties that are common across all sequences regardless of associated user. By contrast, local parameters describe user-specific traits and preferences. They therefore have to be inferred from fewer, user specific data, motivating a Bayesian treatment by the VAE. We further employ a mixture-of-experts approach [Shi et al., 2019] to account for heterogeneity within each user. We demonstrate that our proposed scheme yields systematic improvements in a variety of tasks on four large, real world datasets.

## 2 Personalized Event Sequences

**Problem Statement**  We consider the problem of modeling sequences of events $(t, k)$ that occur at irregular times $t$. Each event carries a mark $k$ corresponding to one of a finite number $K$ of different possible event types. Since all our training sets are finite, each event sequence is bounded by some time horizon $T > 0$, i.e., it can be written as a finite history sequence of the form

$$\mathcal{H}_T = \big((t_1, k_1), (t_2, k_2), \ldots, (t_{|\mathcal{H}_T|}, k_{|\mathcal{H}_T|})\big) \tag{1}$$

with times $0 \leq t_1 < t_2 < \cdots < t_{|\mathcal{H}_T|} \leq T$ and marks $k_i \in \{1, \ldots, K\} \ \forall i$. For brevity, let $\mathcal{H} \equiv \mathcal{H}_T$.

Since our goal is personalization, we assume that each event sequence is associated with a *user*[1] $u$. Our objective is to forecast (or evaluate the likelihood of) an event sequence for a given user $u$ conditioned on three sources of information, illustrated in Figure 1: (i) a large *training set* of event sequences from many users, possibly including $u$ (Figure 1, left); (ii) a smaller (possibly empty) set of *reference sequences* from the same user $u$, that may or may not be part of the training set (blue box in left part of Figure 1); and (iii) a *prefix*, i.e., a (possibly empty) partial sequence $\mathcal{H}_t^u$ of events performed by user $u$ just before the time $t$ where we start predicting (solid circles in right part of Figure 1). Conditioned on these three sources of information, we aim to predict the continuation of the full sequence $\mathcal{H}_{t+\Delta}^u$ (half-transparent circles in Figure 1, right) for some remaining time span $\Delta$.

For example, in an online store, the training set (i) contains many shopping sessions by many users; the reference sequences (ii) are previous sessions of a given user $u$; and the prefix (iii) is the already

observed part of a currently ongoing session. This setup is quite general, covering many applications with recurring users, e.g., social media platforms, music streaming services, or e-commerce websites.

The above enumeration arranges the three types of input information (i)-(iii) as nested subsets organized from global to local context. It thus reflects the assumption that the predicted event sequence $\mathcal{H}^u_{t+\Delta}$ follows some general characteristics (e.g., similarities or incompatibilities between different event types) that can be learned from the entire data set (i). At the same time, we assume that $\mathcal{H}^u_{t+\Delta}$ also exhibits some individual traits of the associated user $u$, which can therefore only be inferred from reference sequences (ii) from the same user. Finally, a user's current goal or mood may still vary somewhat over time and can therefore only be inferred from very recent observations in the prefix $\mathcal{H}^u_t$ (iii).

While the relevance for the current prediction increases as we go from (i) to (iii), the data size shrinks from (i) to (iii). This motivates different treatments of the data sources (i)-(iii) in our models in terms of point estimated global model parameters, a user-specific Bayesian inference variable $z^u$, and the local state of a recurrent neural network, respectively. The rest of this section provides an overview over the proposed framework, deferring a more detailed discussion to Section 3.

**Overview of the Proposed Solution**   Forecasting discrete event sequences amounts to both predicting an ordering of future event *types* $k_i$ as well as their *time stamps* $t_i$. This is a challenging problem due to the strong correlations between these two data modalities. We consider a broad class of stochastic dynamics models—*neural marked temporal point processes* (neural MTPPs)—that autoregressively generate events one after another. For the $i^{\text{th}}$ event, we draw its time stamp $t_i$ and mark $k_i$ conditioned on the hidden state $h_i$ of a "decoder" recurrent neural network (RNN). The RNN state $h_i$ gets updated after each generated event, conditioned on the event it just generated, and on an embedding vector $z^u$ that represents the user $u$. This leads to the following stochastic recursion:

$$(t_{i+1}, k_{i+1}) \sim p_\theta(t_{i+1}, k_{i+1} \mid h_i, z^u)$$
$$\text{with} \quad h_i = f_{\text{Dec}}(h_{i-1}, [\mathbf{k}_i; z^u], t_i; \eta) \quad \text{and} \quad h_0 = \tanh(W_0 z^u + b_0) \tag{2}$$

where $[\,\cdot\,;\,\cdot\,]$ denotes vector concatenation, $\mathbf{k}_i$ is a learnable continuous vector embedding for the mark $k_i$ of the $i^{\text{th}}$ event, and $f_{\text{Dec}}$ is the recurrent unit of the decoder neural network $\text{DEC}_\theta$ with learnable parameters $\theta$ that include $\eta$, $W_0$, and $b_0$.

Apart from the user embedding $z^u$, which is specific to our personalization scheme, Eq. 2 covers several MTPP models in the literature. Section 3 summarizes models that fit into our framework. Note that we are not considering any side-information from users (e.g., age, location, gender, etc.); however, depending on the application these should be straightforward to incorporate into the framework.

Eq. 2 models the generation of event sequences probabilistically: each successive event $(t_{i+1}, k_{i+1})$ is sampled from $p_\theta(t_{i+1}, k_{i+1} \mid h_i, z^u)$ rather than being generated deterministically. Probabilistic models allow for the ability to estimate complex statistics, such as the expected time until next event of a specific type, how much more likely event A will come before event B, etc. Many of these questions would be hard or impossible to answer with a deterministic model.

Eq. 2 further reflects our problem setting with diverse information sources (i)-(iii) discussed above. The decoder parameters $\theta$ are identical for all generated sequences and can thus be trained on the entire data set (i). The user embedding $z^u$ stays constant within each event sequence but varies from user to user. Thus, $z^u$ has to be inferred from reference sequences (ii) from the same user. By concatenating the user embedding to the mark embedding, we effectively create personalized representations of events for that user. Additionally, by computing the initial stochastic state $h_0$ from $z^u$, we allow for personalized predictions across the entire time window from $t = 0$ to $t = T$. Finally, when completing a partial sequence, the prefix (iii) can be encoded into the initial RNN state $h_i$ by unrolling the RNN update (second line of Eq. 2) on the events in the prefix.

## 3   Model Parameterization and Inference

This section specifies the details for representing and estimating user embedding $z^u$, the underlying data generating process for sequential event data, how existing neural MTPP models can be extended to become personalized by incorporating user embeddings, as well as the loss function being optimized. Figure 2 shows operational diagrams of the encoding and decoding processes.

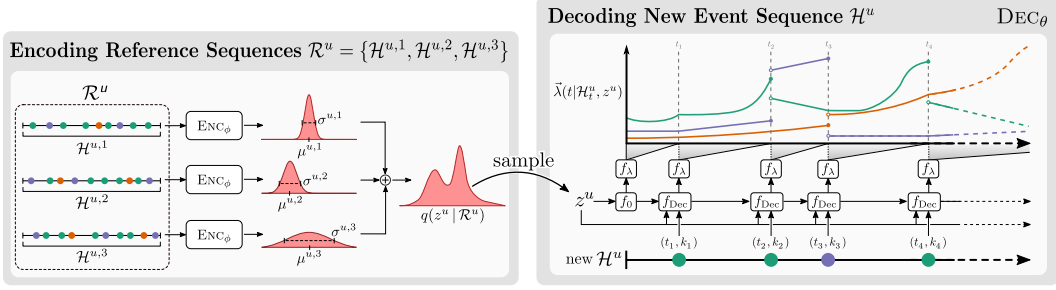

Figure 2: On the left is the encoding process for three reference sequences from user $u$ that belong to the reference set $\mathcal{R}^u$. This results in the approximate posterior mixture distribution $q(z^u \,|\, \mathcal{R}^u)$ that is then sampled and used in the decoding process on the right for a target sequence $\mathcal{H}^u$. $f_\lambda$ is the model specific transformation of hidden state $h_i$ and time $t \in (t_i, t_{i+1}]$ to intensity rates across marks.

**Encoding User Embeddings** The user embedding $z^u$ in Eq. 2 allows us to personalize predictions for a given user $u$. $z^u$ is a real-valued vector, and for our results the dimensionality ranges from 32 to 64 depending on the dataset (see supplement for more information). The vector $z^u$ can be interpreted as the sequence and user-specific dynamics for a single history of events. We infer $z^u$ from a reference set $\mathcal{R}^u = \{\mathcal{H}^{u,1}, \ldots, \mathcal{H}^{u,n^u}\}$ of $n^u$ sequences that we have already observed from the same user. This leads to two complications: first, the amount of data in each reference set $\mathcal{R}^u$ is much smaller than the whole training set of sequences from all users; second, learning an individual user embedding $z^u$ for thousands of users would be expensive. We address both complications by an approximate Bayesian treatment of $z^u$ via amortized variational inference (amortized VI) [Kingma and Welling, 2014, Rezende et al., 2014, Zhang et al., 2019a].

The typically small amount of data in each reference set $\mathcal{R}^u$ motivates a Bayesian treatment via a probabilistic generative process with a prior $p(z^u)$ and the likelihood in Eq. 2. For simplicity, we assume a standard normal prior: $z^u \sim \mathcal{N}(0, I)$. Bayesian inference seeks the posterior probability $p(z^u|\mathcal{R}^u)$. As finding the true posterior is computationally infeasible, VI approximates the posterior with a parameterized variational distribution $q(z^u|\mathcal{R}^u)$ by minimizing the Kullback-Leibler divergence from the true posterior to $q$. The inferred approximate posterior $q(z^u|\mathcal{R}^u)$ can then be used to sample a user embedding $z^u$ for a personalized prediction, i.e.,

$$z^u \sim \begin{cases} \mathcal{N}(0, I) & \text{for unconditional generation;} \\ q(z^u \,|\, \mathcal{R}^u) & \text{for personalized prediction.} \end{cases} \tag{3}$$

In principle, one could fit an individual variational distribution $q(z^u|\mathcal{R}^u)$ for each user $u$. As this would be expensive, we instead use amortized VI [Kingma and Welling, 2014] We first model $q(z^u|\mathcal{R}^u)$ by a mixture of experts [Shi et al., 2019] where each expert $q(z^u|\mathcal{H}^u)$ is conditioned only on a single reference sequence $\mathcal{H}^u \in \mathcal{R}^u$,

$$q(z^u \,|\, \mathcal{R}^u) = \frac{1}{n^u} \sum_{i=1}^{n^u} q(z^u \,|\, \mathcal{H}^{u,i}). \tag{4}$$

$q(z^u \,|\, \mathcal{R}^u)$ represents the various modes of dynamics for a given user $u$ as defined by their past sequences $\mathcal{R}^u$. Further, each expert distribution $q(z^u|\mathcal{H}^u)$ is a fully factorized normal distribution where the means $\mu$ and variances $\sigma^2$ are further parameterized by an encoder neural network $\text{ENC}_\phi$ with parameters $\phi$ that are shared across all users,

$$q(z^u \,|\, \mathcal{H}^u) = \mathcal{N}\big(z^u; \mu, \text{diag}(\sigma^2)\big) \qquad \text{where} \qquad (\mu, \log \sigma) = \text{ENC}_\phi(\mathcal{H}^u). \tag{5}$$

$\text{ENC}_\phi$ contains a bidirectional RNN (more specifically gated recurrent units) that takes embedded times and marks of the reference sequence as inputs. The mark embeddings are learned and the time embeddings are continuous versions of the fixed positional embeddings [Cho et al., 2014, Vaswani et al., 2017]. The last hidden states from both directions are concatenated and then linearly transformed to result in $\mu$ and $\log \sigma$. Precise details for this process can be found in the supplement.

Eqs. 4-5 specify the variational family. We optimize over the variational parameters $\phi$ using standard methods for Black Box VI [Blei et al., 2017, Zhang et al., 2019a], i.e., by stochastic maximization of the evidence lower bound.

**Distributions for Events**   Marked temporal point processes (MTPP) are a broad class of processes used for modeling seqeunces of events $\mathcal{H}$. A common method to fully characterize a MTPP is through an intensity function,

$$\lambda(t|\mathcal{H}_t) = \lim_{\delta \downarrow 0} \tfrac{1}{\delta} P\big(|\mathcal{H}_{t+\delta}| - |\mathcal{H}_t| = 1 \mid \mathcal{H}_t\big), \tag{6}$$

where $|\mathcal{H}_t|$ counts the number of events up to time $t$. The intensity function measures the instantaneous rate of occurrence for events at time $t$, conditioned on the history up until time $t$. Mark-specific intensity functions are defined as the product of the overall intensity function and the conditional categorical distribution over marks, $\lambda_k(t|\mathcal{H}_t) = p(k|t, \mathcal{H}_t)\lambda(t|\mathcal{H}_t)$. We denote the vector of rates over all marks as $\vec{\lambda}(t|\mathcal{H}_t)$. The log-likelihood of a sequence $\mathcal{H}$ works out to be

$$\log p(\mathcal{H}) = \sum_{i=1}^{|\mathcal{H}|} \log \lambda_{k_i}(t_i|\mathcal{H}_{t_i}) - \int_0^T \lambda(t|\mathcal{H}_t)dt. \tag{7}$$

Intuitively, the summation in the first term rewards the model when the intensity values are high for the actual observed marks within the history, whereas the negative integral term penalizes high overall intensity values when there is no event.

Intensity functions have been parameterized in both simple forms for interpretability as well as with neural networks for flexibility. Our proposed approach can in principle be used to add personalization to most existing neural MTPP models. We selected two of the most well-known and widely-cited such models to serve as base architectures which we extend for personalization as described in Sec. 2:

- The *Recurrent Marked Temporal Point Process* (RMTPP) [Du et al., 2016], which parameterizes the intensity function explicitly as a piece-wise exponentially decaying rate: $\vec{\lambda}^{\text{RMTPP}}(t|\mathcal{H}_t) = \exp\{Wh_i + w(t - t_i) + b\}$, where $t_i < t \leq t_{i+1}$, and $W$, $w$, and $b$ are learnable parameters. For this model, $f_{\text{Dec}}$ from Eq. 2 is a gated recurrent unit (GRU).
- The *Neural Hawkes Process* (NHP) [Mei and Eisner, 2017], which describes a procedure to obtain an interpolated hidden state $h(t)$, defines $\vec{\lambda}^{\text{NHP}}(t|\mathcal{H}_t) = \text{softplus}(Wh_i(t))$ where $W$ is a learnable matrix with $t_i < t \leq t_{i+1}$. and has $f_{\text{Dec}}$ from Eq. 2 be a continuous-time LSTM unit.

By incorporating $z^u$ as specified in Eq. 2, $\vec{\lambda}(t|\mathcal{H}_t^u)$ becomes $\vec{\lambda}(t|\mathcal{H}_t^u, z^u)$. Note that by defining $\vec{\lambda}(t|\mathcal{H}_t^u, z^u)$ and $f_{\text{Dec}}$, we effectively define $p(t_{i+1}, k_{i+1} \mid h_i, z^u)$ in Eq. 2, as well as the general decoding process, $\text{DEC}_\theta$.

All MTPP models can be sampled via a thinning procedure [Ogata, 1981], if not directly. Similarly, the integral in Eq. 7 can be computed by Monte-Carlo estimation, if not analytically. In our experiments, we perform the former for all models for consistency. More precise details on this can be found in the supplement.

**Optimization**   The objective function for the proposed personalized neural MTPP models is the $\beta - $ VAE objective [Higgins et al., 2016] which is defined for a single target sequence $\mathcal{H}^u$ as:

$$\mathcal{L}_\beta(\phi, \theta; \mathcal{H}^u) = \mathbb{E}_{q_\phi(z^u|\mathcal{R}^u)}[\log p_\theta(\mathcal{H}^u|z^u)] - \beta \text{KL}(q_\phi(z^u|\mathcal{R}^u)||p(z^u)), \tag{8}$$

which the right-hand side is a variant of what is known as the evidence lower bound (ELBO). The expectation is estimated with a Monte-Carlo estimate. During training, a single sample for the estimate turned out to be sufficient, whereas during testing we utilized 5 samples to reduce variance.

## 4   Experimental Results

To measure the effectiveness of this framework in general, using the NHP and RMTPP base models we trained each in their standard configuration (i.e., a decoder-only setup) and in the proposed variational mixture-of-experts setup (referred to as MoE-NHP and MoE-RMTPP).[2] We rated these models on their held-out log-likelihood values, next event predictions, and user/source identification capabilities as described below.[3] Furthermore, all tests were conducted using sequences from new users to emphasize the added ability to adapt to new sources.

Table 1: Statistics for the four datasets. Columns (left to right) are: total time window $T$ for every sequence; number $K$ of unique marks; mean sequence length $|\mathcal{H}|$; mean number $|\mathcal{R}^u|$ of sequences per user; total number of sequences and number of unique users in training/validation/test splits.

| Dataset | $T$ | $K$ | Mean $|\mathcal{H}|$ | Mean $|\mathcal{R}^u|$ | # Sequences | | | # Unique Users | | |
|---|---|---|---|---|---|---|---|---|---|---|
| | | | | | Train | Valid | Test | Train | Valid | Test |
| Meme | 1 Week | 5000 | 23.4 | 6.9 | 271K | 9K | 21K | 31K | 1K | 3K |
| Reddit | 1 Week | 1000 | 65.2 | 4.9 | 343K | 15K | 34K | 49K | 3K | 7K |
| Amazon | 1 Month | 737 | 10.7 | 4.0 | 262K | 8K | 20K | 28K | 2K | 5K |
| LastFM | 1 Day | 15 | 45.6 | 347 | 289K | 15K | 49K | 833 | 44 | 105 |

Models were trained by minimizing Eq. 7 and Eq. 8, averaged over training sequences, for the decoder-only and MoE variants respectively via the Adam optimizer with default hyperparameters [Kingma and Ba, 2014] and a learning rate of 0.001. A linear warm-up schedule for the learning rate over the first training epoch was used as it led to more stable training across runs. We also performed cyclical annealing on $\beta$ in Eq. 8 from 0 to 0.001 with a period of 20% of an epoch to help prevent the posterior distribution from collapsing to the prior [Fu et al., 2019].

### 4.1 Datasets

All models were trained and evaluated on four real-world datasets (see Table 1). The **MemeTracker** dataset [Leskovec and Krevl, 2014] relates to common phrases (memes). We defined the meme as the "user" and the website it was posted to as the mark. The mark vocabulary is the set of top 5000 websites by volume of utterances. Sequences were defined as one-week-long chunks spanning August 2008 to April 2009, and event times were measured in hours. The **Reddit comments** dataset [Baumgartner et al., 2020] relates to user-comments on posts in the social media site `reddit.com`. One month of data (October 2018) was used to extract user sequences, and the mark vocabulary was defined as the top 1000 communities (subreddits) by comment volume. The month was divided into multiple sequences consisting of week-long windows per user, with event times in units of hours. **Amazon Reviews** [Ni et al., 2019] consists of timestamped product reviews between May 1996 and October 2018, with marks defined as 737 different product categories. User sequences were defined as 4-week windows with event times in units of days (with a small amount of uniformly distributed noise to avoid having multiple events co-occur). The 4th dataset, **LastFM** [Celma, 2010], has time-stamped records of songs listened to (both artists and track names) for nearly 1,000 users on the `last.fm` website. Marks were defined as one of 15 possible genres for a song, via the `discogs.com` API. User sequences corresponded to 1 day of listening in units of hours. For all datasets event times were calculated relative to the start-time of a user sequence. All datasets were filtered to include sequences with at least five events and no more than 200. Training, validation, and test sets were split so that there were no users in common between them.

### 4.2 Results

**Training Data Size Ablation**   We first investigate differences in predictive performance between the two proposed Mixture of Experts (MoE) models and their decoder-only counterparts, as a function of the size of the training data. We therefore trained each model on various subsets of the training data, using 10%, 20%, 30%, 50%, 70%, 90%, and 100% of the full training set. For efficiency reasons, we generated these trained models using curriculum learning, i.e., we first trained each model on the 10% subset until convergence, then added 10% more training points and trained on the resulting 20% subset, and so on. Convergence on each subset was determined when validation log-likelihood improved by less than 0.1. The training subsets were generated via random sampling of users. All models were evaluated on the same fixed-size test dataset.

The results can be seen in Figure 3(a). The proposed MoE models (solid lines) systematically yield better predictions in terms of test log-likelihood over their non-MoE counterparts (dotted lines). This trend suggests that our personalization scheme could benefit most, if not all, neural MTPP models and that these benefits appear for even small amounts of training data.

**Likelihood Over Time**   Having seen that the proposed MoE models have better predictive log-likelihoods than their decoder-only counterparts, we now investigate where exactly they achieve these

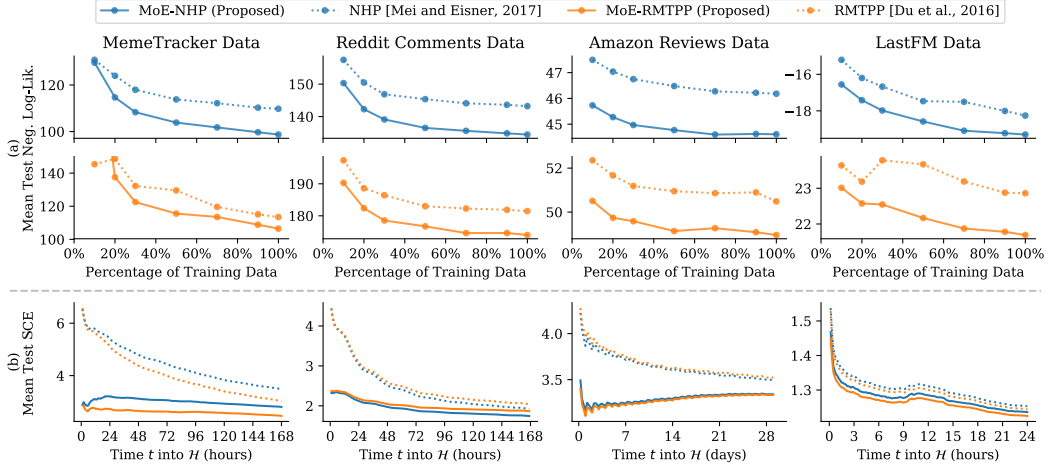

Figure 3: (a) Mean test negative log-likelihood performance for models trained at varying percentages of the training data across the four datasets and (b) mean test cross entropy up to time $t$ (see Eq. 9) for models trained on 100% of the training data (lower is better for both). Results for NHP-based models are shown in blue, RMTPP models are orange, decoder-only models are dotted lines, and the proposed mixture-of-experts models are solid lines.

performance gains. Using $\lambda_k(t|\mathcal{H}_t) = p(k|t, \mathcal{H}_t)\lambda(t|\mathcal{H}_t)$, we can factor Eq. 7 as follows:

$$-\log p(\mathcal{H}) = |\mathcal{H}| \left(\mathrm{SCE}(\mathcal{H}) + \mathrm{PP}^+(\mathcal{H})\right) + T\,\mathrm{PP}^-(\mathcal{H}) \tag{9}$$

where $\mathrm{SCE}(\mathcal{H}_t) \equiv \frac{-1}{|\mathcal{H}_t|} \sum_{i=1}^{|\mathcal{H}_t|} \log p(k_i|t_i, \mathcal{H}_{t_i})$, is the average cross entropy of a sequential, non-continuous time based, classification model, and $\mathrm{PP}^+(\mathcal{H}_t) \equiv \frac{-1}{|\mathcal{H}_t|} \sum_{i=1}^{|\mathcal{H}_t|} \log \lambda(t_i|\mathcal{H}_{t_i})$, $\mathrm{PP}^-(\mathcal{H}_t) \equiv \frac{1}{t} \int_0^t \lambda(\tau|\mathcal{H}_\tau)d\tau$ respectively represent the average positive and negative evidence of a sequence, ignoring the associated marks (the two together make up the terms in the log-likelihood of a non-marked temporal point process). All of these terms can assist in identifying issues within MTPPs, especially when investigated as a function of $t$. We presume that most of the heterogeneity between sequences in datasets resides in the categorical distributions of marks. Here we focus our analysis on the SCE term—see the supplementary material for discussion of other terms.

Figure 3(b) shows average SCE values over time. The decoder-only models (dotted) tend to have high SCE values near the beginning of sequences (left side of x-axis), as the model adapts to the type of sequence and user it is making predictions for. In contrast, the MoE models (solid) have much lower SCE near the beginning of sequences, i.e., are making significantly better categorical predictions for marks early on. The decoder-only models gradually approach the performance of the MoE models over time, but never close the gap, indicating the user information (via $z_u$) provides significant benefit in mark prediction that is difficult for an RNN decoder model to learn from the sequence itself.

**Ranking of Next Event Predictions**    One use case of MTPP models is to predict what a user will do next and when they will do it during an ongoing sequence. As an example scenario, we conditioned the models on a prefix $\mathcal{H}_{t_{10}}^u$ of the first 10 events in each test sequence and evaluated the predictive performance for the next event $(t_{11}, k_{11})$. Predicted times and marks were estimated by marginalizing over marks and times respectively to minimize Bayes risk, similar to Mei and Eisner [2017]—see the supplement for details. The choice of 10 events in the prefix was made to simulate making predictions early in a sequence, where there is still a good deal of variability for the next event.

The predicted marks were evaluated by the ranking of the true mark's predicted probability, averaged over all sequences in the test set for each dataset. Table 2(a) shows our results. MoE models achieve superior performance than the other models on all datasets, with particularly large gains for the Meme and Reddit data. The results for predicted times were not found to be consistently different between the MoE and non-personalized models. One possible reason for this might be that there is not a strong user-specific signal in event timing information that cannot already be detected in the sequence being

Table 2: (a) Mean predicted mark rank for the proposed MoE-NHP and MoE-RMTPP models versus the non-personalized NHP and RMTPP models. More precision is shown for datasets with fewer total marks. (b) Source identification error rates for MoE-NHP, MoE-RMTPP, and Bayesian Poisson process baseline across the four datasets. For both, lower percentages are better and the best performance within a grouping is bolded. In all categories, one of our proposed models is the best.

| | (a) Mean Predicted Rank ($\hat{k}_{11}$) | | | | | (b) Source Identification Error Rate | | |
|---------|---------|------|-----------|-------|---------|---------|------------|----------|
| Dataset | MoE-NHP | NHP | MoE-RMTPP | RMTPP | Dataset | MoE-NHP | MoE-RMTPP | Bayes PP |
| Meme | **225** | 283 | **143** | 159 | Meme | 5.04% | **4.64%** | 9.18% |
| Reddit | **19.4** | 35.2 | **20.0** | 35.6 | Reddit | **1.38%** | 2.02% | 3.24% |
| Amazon | **21.6** | 22.8 | **21.6** | 22.9 | Amazon | **8.34%** | 9.94% | 12.88% |
| LastFM | **2.02** | 2.04 | **2.01** | 2.04 | LastFM | **28.44%** | 33.90% | 39.94% |

decoded, thus causing the personalization models to focus more on event types than on times. In terms of predictive performance, it appears that the biggest benefit of personalization is more accurate predictions of marks rather than times. All results for predicted times can be found in the supplement.

**Source Identification**   The personalized MoE models naturally lend themselves to being able to detect anomalous events for a given user. We evaluate how well the models can identify the source of a given event sequence. As discussed in Sec. 2, the MoE models were designed such that conditioning on a user embedding, subsequent sequences from the same user would be more likely (hence being personalized), and in turn sequences from different users would be less likely. We assessed this behavior on all four datasets by randomly selecting target sequences and pairing them up with one reference sequence from the same user and one reference sequence from a different user. After conditioning on the reference sequences, the models evaluated the likelihood of the target sequence and the two results were ranked amongst each other. This was done 5,000 times for each test dataset (10,000 total likelihood calculations each). The target sequences were truncated to 10 events in order to simulate identifying users from only a portion of events into a session.

The MoE-NHP and MoE-RMTPP models were compared against a Bayesian Poisson process baseline with a Gamma prior where the prior expected value was determined by the MLE of the training data and the strength of the prior was tuned on the validation set. Each target sequence was then evaluated on the posterior distribution conditioned on the reference sequence—more details can be found in the supplement. The results in Table 2(b) show that both of the MoE models have significantly lower error rates than the baseline on this task, across all four datasets.

## 5   Related Work

There has been a significant amount of recent work on neural MTPPs that has focused on incorporating additional information outside of the event sequence itself, such as dense time-series information [Xiao et al., 2017b]. There have also been advances in bypassing computational issues involved in evaluating integrals in the MTPP likelihood, e.g., via direct modeling of cumulative hazard functions [Omi et al., 2019] or via the use of normalizing flows [Shchur et al., 2020]. Other approaches have avoided these issues through alternative objectives à la reinforcement learning [Li et al., 2018, Upadhyay et al., 2018, Zhu et al., 2020] or adversarial training [Xiao et al., 2018, 2017a].

In the context of user-specific sequence models, Vassøy et al. [2019] introduced MTPPs with learned user embeddings for applications with many sequences per user to personalize next step recommendations. In contrast, our work focuses on forecasting entire future trajectories of user activity in situations with few sequences per user. Additionally, we concern ourselves with predicting for users that were not necessarily present during model fitting.

Sequential VAE models [Bayer and Osendorfer, 2014, Chung et al., 2015, Gan et al., 2015] have been developed for discrete-time and dense data such as text [Bowman et al., 2016, Yang et al., 2017, Schmidt et al., 2019] or video [Li and Mandt, 2018, Denton and Fergus, 2018] with applications in compression [Lombardo et al., 2019] and control [Chua et al., 2018, Ha and Schmidhuber, 2018]. However, none of these works have been extended to the continuous-time and sparse data domain that we encounter in event sequence modeling.

# 6    Conclusion

We address the problem of personalization in discrete-event continuous time sequences by proposing a new framework that integrates with existing neural MTPP models based on a variational mixture-of-experts autoencoder approach. Our analysis shows that, across four different real world datasets, the introduction of personalization via our proposed framework improved predictive performance on metrics such as held-out log-likelihood, next event prediction, and source identification. Our approach opens up possibilities for various personalized tasks that were not readily feasible before with prior neural MTPP works. Such tasks include personalized sequential recommendations, demand forecasting, and source-wise anomaly detection.

It should be noted, however, that the fact that the $\beta$-VAE approach, with $\beta < 1$, was necessary showed that our approach suffers from similar problems as sequential VAEs for text generation Bowman et al. [2016]. This makes the learned representations less useful in practice and opens up the question on which models are better suited for representation learning of discrete event sequences. We foresee future work tackling this issue through potentially several avenues: better model architectures (for either the encoder or decoder), better multi-modal latent modeling mechanisms instead of the mixture-of-experts approach, or even more appropriate priors that better match the underlying sequence dynamics.

## Acknowledgements

This material is based upon work supported in part by the National Science Foundation under grant numbers 1633631, 1928718, 2003237, and 2007719; by the National Science Foundation Graduate Research Fellowship under grant number DGE-1839285; by the Center for Statistics and Applications in Forensic Evidence (CSAFE) through Cooperative Agreement 70NANB20H019 between NIST and Iowa State University, which includes activities carried out at University of California Irvine; by the Defense Advanced Research Projects Agency (DARPA) under Contract No. HR001120C0021; by an Intel grant; and by a Qualcomm Faculty Award. Any opinion, findings, and conclusions or recommendations expressed in this material are those of the authors and do not necessarily reflect the views of the National Science Foundation, nor do they reflect the views of DARPA.

Additional revenues related to this work include: employment with the Walt Disney Company and Google; research funding from NIH, NASA, NIST, PCORI, SAP and eBay; honoraria from General Motors; consulting income from Toshiba; and internships at Workday Incorporated, NVIDIA, and Microsoft Research.

## Broader Impacts

While many of the successful and highly-visible applications of machine learning are in classification and regression, there are a broad range of applications that don't naturally fit into these categories and that can potentially benefit significantly from machine learning approaches. In particular, in this paper we focus on continuous-time event data, which is very common in real-world applications but has not yet seen significant attention from the ML research community. There are multiple important problems in society where such data is common and that could benefit from the development of better predictive and simulation, including:

- **Education:** Understanding of individual learning habits of students, especially in online educational programs, could improve and allow for more personalized curricula.
- **Medicine:** Customized tracking and predictions of medical events could save lives and improve patients' quality of living.
- **Behavioral Models:** Person-specific simulations of their behavior can lead to better systematic understandings of people's social activities and actions in day-to-day lives.
- **Cybersecurity:** Through the user identification capabilities, our work could aid in cyber-security applications for the purposes of identifying fraud detection and identify theft.

Another potential positive broad impact of the work, is that by utilizing amortized VI, our methods do not require further costly training or fine-tuning to accommodate new users, which can potentially produce energy savings and lessen environmental impact in a production setting.

On the other hand, as with many machine learning technologies, there is also always the potential for negative impact from a societal perspective. For example, more accurate individualized models for user-generated data could be used in a negative fashion for applications such as surveillance (e.g., to monitor and negatively impact individuals in protected groups). In addition, better predictions and recommendations for products and services, through explicitly conditioning on prior behavior from a user, could potentially further worsen existing privacy concerns.

## Footnotes

[1]More generally, "user" denotes any source of event sequences, e.g., an individual, organization, or system.

[2]Our source code for modeling and experiments can be found at the following repository: `https://github.com/ajboyd2/vae_mpp`.

[3]Sampling quality was also evaluated. Details and results can be found in the Supplement.

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
