[Supplementary Material]

# Supplement to: User-Dependent Neural Sequence Models for Continuous-Time Event Data

## A  Model Details

### A.1  Encoder Details

Below are precise details concerning the steps to encode a single sequence, $H^u$, to describe its associated component distribution, $q(z^u \mid \mathcal{H}^u)$.

**Temporal Embedding**  The encoder contains a bidirectional RNN that accepts as input for each step the embedded mark vector, $\mathbf{k}_i$, and the embedded time vector, $\mathbf{t}_i$, for a given event in the sequence being encoded. The embedding for the mark is a standard, learned embedding. The embedding for the time is a fixed transformation that converts a single time $t$ into a $d_{\text{time}}$-dimensional vector. The specific temporal embedding function used in the models trained is defined as follows:

$$\Phi(t) = [\sin(\alpha(t - t_i)); \cos(\alpha(t - t_i))]$$
$$\text{for } t_i < t \le t_{i+1}, \tag{A}$$

where $t_i$ is the latest event time that the model has conditioned on that is less than the time $t$ being embedded, $\alpha$ is a fixed $\frac{d_{\text{time}}}{2}$-dimensional vector with the $j$th element being $\alpha_j = \exp\{-j \log(T_{max})/d_{\text{time}}\}$ with $T_{max}$ being the maximum difference of consecutive times for a given dataset. This transformation is the same as the positional embeddings from Vaswani et al. [2017] adapted to continuous times and can be seen as a simplified version of Xu et al. [2019]. This form was chosen to have a dense representation of time that safely generalizes to new time values.

**Encoding Events**  As mentioned previously, the embedded times and marks are used as inputs to a bidirectional RNN. More precisely:

$$\overrightarrow{h}_i = f_{\overrightarrow{\text{Enc}}}(\overrightarrow{h}_{i-1}, [\mathbf{t}_i; \mathbf{k}_i]), \tag{B}$$

$$\overleftarrow{h}_i = f_{\overleftarrow{\text{Enc}}}(\overleftarrow{h}_{i+1}, [\mathbf{t}_i; \mathbf{k}_i]), \tag{C}$$

for $i = 1, \dots, |\mathcal{H}^u|$ where $\overrightarrow{h}_0$ and $\overleftarrow{h}_{|\mathcal{H}^u|+1}$ are learned, and $f_{\overrightarrow{\text{Enc}}}$ and $f_{\overleftarrow{\text{Enc}}}$ are recurrent units (in our case, GRUs).

The information in the reference sequence is summarized by concatenating the last hidden states from each direction. We will denote that as $h = [\overrightarrow{h}_{|\mathcal{H}^u|}; \overleftarrow{h}_1]$. This vector is then used to compute the sufficient statistics for the component distribution via:

$$\mu = f_\mu(h) \quad \text{and} \quad \log \sigma = f_\sigma(h) \tag{D}$$

where in our implementation, $f_\mu(h) = W_\mu h + b_\mu$ and $f_\sigma(h) = W_\sigma h + b_\sigma$ for learnable matrices $W_\mu$ and $W_\sigma$ and learnable bias vectors $b_\mu$ and $b_\sigma$.

Figure A: A detailed operational diagram of the encoding process. A single sequence $\mathcal{H}^u$ is being encoded to compute the sufficient statistics, $\mu^u$ and $\sigma^u$, for a single component/mixture, $q(z^u \mid \mathcal{H}^u)$.

Figure B: Graphical model of the proposed VAE for personalized point processes with a) being the generative model and b) being the inference model. $\mathcal{U}$ is the set of all possible users for a given dataset.

## A.2 Model Visualizations

An operational diagram of the encoder, as described previously, can be found in Figure A. A graphical model representation of the personalized neural MTPP framework can be found in Figure B.

## A.3 Sampling Sequences

One convenient property of point processes is that given multiple point processes, the superposition of them is also a valid point process. Furthermore, the intensity function of the superposition point process is simply equal to the sum of intensity functions that make it up (e.g., for the combination of two point processes: $\lambda(t|\mathcal{H}_t) = \lambda_1(t|\mathcal{H}_t) + \lambda_2(t|\mathcal{H}_t)$). This is why $\sum_{k=1}^{K} \lambda_k(t|\mathcal{H}_t) = \lambda(t|\mathcal{H}_t)$ for MTPPs, as events for each mark can be seen as coming from their own point process. Note that no assumption of independence was made for this property to hold true.

We utilize this property in a thinning procedure to sample from arbitrary point processes. If we let $\lambda^* > \lambda(t|\mathcal{H}_t)$ for all $t \in [0, T)$, then to sample from $\lambda(t|\mathcal{H}_t)$ requires sampling a time, $t^*$, from a Poisson point process with constant rate $\lambda^*$, then randomly accepting that point as originating from the model with probability $\frac{\lambda(t^*|\mathcal{H}_t)}{\lambda^*}$. If it is accepted, then the mark is determined by sampling from a

Table A: Descriptions and values of hyperparameters used for models trained on all of the datasets. The same decoder hyperparameters were used on all of the models, whereas the encoder hyperparameters were only used for the MoE variants.

| Section | Description | Value Used | | | |
|---------|-------------|------|--------|--------|--------|
|         |             | Meme | Reddit | Amazon | LastFM |
| Encoder | Temporal Embedding Size | 64 | 64 | 64 | 64 |
|         | Mark Embedding Size | 32 | 32 | 32 | 32 |
|         | Encoder Hidden Size | 64 | 64 | 64 | 64 |
|         | Latent State Size | 64 | 32 | 32 | 32 |
| Decoder | Mark Embedding Size | 32 | 32 | 32 | 32 |
|         | Decoder Hidden Size | 64 | 64 | 64 | 256 |

categorical distribution with probabilities equal to $\frac{\lambda_k(t^*|\mathcal{H}_t)}{\lambda(t^*|\mathcal{H}_t)}$ for $k \in \{1, \ldots, K\}$. The time and mark are then appended to $\mathcal{H}$ and the procedure continues until candidate times are sampled outside of a pre-specified time window. Note that, should we want to condition on a portion of a history, $\mathcal{H}_c$, and sample future trajectories, then the intensity function is conditioned on $\mathcal{H}_c$ and the only candidate times considered are $t \in [c, T)$.

For this thinning procedure to be valid, $\lambda^*$ must dominate all intensity values estimated by the model; however, this can be difficult to ensure prior to generating a sample due to the intensity function changing in response to new events. As such, to ensure that a sample was validly generated one must check the intensity function conditioned on the sample at various times in the time window and validate that it is less than $\lambda^*$. All samples that we generated had this check done at 1,000 different times that were uniformly sampled across the time window. Should this condition not hold for at least one point in the interval, then $\lambda^*$ is increased and a new sample is generated and subsequently validated.

## B  Experimental Details

This section pertains to more precise and specific details concerning our experimental findings, as well as some additional results that were omitted from the main text due to space.

### B.1  Experiment Hyperparameters

Below are descriptions that list all of the hyperparameters set throughout our training and experiments, such as the specific sizes for model parameters or the number of samples used when approximating integrals.

**Model Architecture**  Table A contains model hyperparameters used for all of the experiments. The same hyperparameters were sufficient for models trained on all datasets; however, due to the difference in total unique marks (i.e. $K_{\text{MemeTracker}} = 5000$, $K_{\text{Reddit}} = 1000$, $K_{\text{Amazon}} = 737$, and $K_{\text{LastFM}} = 15$) it turned out necessary to have the size of the amortized latent user embedding, $z_u$, be 64 for the MemeTracker data instead of 32 as used for the Reddit and Amazon data. Similarly, we found it necessary that the decoder hidden size was much larger for the models trained on the LastFM data due to different trends and patterns being harder to differentiate from the subset of marks present alone. These values were found from ablation studies using splits of the training data for each dataset. We suspect that since each dataset had a similar amount of observations, the models tended to require similar minimum capacities.

For the MoE models, to save on parameters, the same mark embeddings were shared amongst the encoder and decoder.

**Approximations**  For training and experiments, there are a number of integrals that need to be computed which are not feasible in closed form. Thus, we must approximate them. All integrals and expectations are approximated via Monte-Carlo (MC) estimates with varying amounts of samples used.

Figure C: Mean values of SCE, PP$^+$, and PP$^-$ (seen in rows (a), (b), and (c), respectively) for decoder-only (dotted lines) and MoE (solid lines) variants of NHP (blue lines) and RMTPP (orange lines) models with each column corresponding to results for the MemeTracker dataset, Reddit comments, Amazon reviews, and the LastFM dataset. In every plot, a lower value is better. Note that for the Amazon results, the periodic trends shown can most likely be attributed to the small uniform noise applied to the data to avoid multiple events co-occurring.

In the log-likelihood of a sequence, Eq. 7, the term $\int_0^T \lambda(t|\mathcal{H}_t)dt$ uses 150 MC samples during training and 500 MC samples for evaluating held-out log-likelihood values for experiments. The exact approximation procedure for the log-likelihood can be found in Mei and Eisner [2017]. Similarly, in the objective function for the MoE models, Eq. 8, the expectation term $\mathbb{E}_{q_\phi(z^u|\mathcal{R}^u)}[\log p_\theta(\mathcal{H}^u|z^u)]$ uses a single MC sample drawn from $q_\phi(z^u|\mathcal{R}^u)$ for training, and 5 MC samples for evaluating held-out log-likelihood values for experiments.

When evaluating the integrals used for next event predictions, we used 10,000 samples where the sample points were shared across integrals for a single set of predictions in order to save on computation. This approach is the same as executed in Mei and Eisner [2017].

## B.2 Likelihood Over Time Analysis

In the main text, we broke down the negative log-likelihood of a sequence $\mathcal{H}$ up to time $t$ into the normalized cross entropy $\mathrm{SCE}(\mathcal{H}_t) = \frac{-1}{|\mathcal{H}_t|} \sum_{i=1}^{|\mathcal{H}_t|} \log p(k_i|t_i, \mathcal{H}_{t_i})$, the normalized positive point process evidence $\mathrm{PP}^+(\mathcal{H}_t) = \frac{-1}{|\mathcal{H}_t|} \sum_{i=1}^{|\mathcal{H}_t|} \log \lambda(t_i|\mathcal{H}_{t_i})$, and the normalized negative point process evidence $\mathrm{PP}^-(\mathcal{H}_t) = \frac{1}{t} \int_0^t \lambda(\tau|\mathcal{H}_\tau)d\tau$. The first of which assesses the model's sequential classification performance and the latter two assesses the model's performances as a non-marked temporal point process (or in other words, how well it captures the time dynamics of the event data). More specifically, the positive evidence measures how well the model reports high intensity rates for all events when an event actually occurs, and the negative evidence quantifies how well the model estimates low intensity values during periods of no events occurring. These defined terms have all been normalized so that they may be compared across different amounts of time into a sequence. They have also been appropriately negated so that for each term, a lower value is desirable.

Results for all three terms across all four model variants and all four datasets can be seen in Figure C. We observe that the cross entropy for every dataset is superior for the proposed, personalized models

Table B: Mean next event prediction results for the predicted times $\hat{t}_{11}$ after conditioning on ten events in a given sequence. $L_1$ error is reported comparing true times to predicted times, with lower values being better. Superior performance between an MoE model and decoder-only counterpart is bolded for every dataset.

| Dataset | Mean L1 Loss for Predicted Times ($\hat{t}_{11}$) | | | |
| | MoE-NHP | NHP | MoE-RMTPP | RMTPP |
| --- | --- | --- | --- | --- |
| Meme | **15.64** | 15.93 | **12.97** | 14.01 |
| Reddit | 4.57 | **4.51** | 3.88 | **3.87** |
| Amazon | **2.39** | 2.43 | **1.97** | 2.06 |
| LastFM | **0.56** | 0.61 | 0.33 | **0.32** |

compared to their decoder-only counterparts, especially near the beginning of the sequence where there is the most uncertainty. The story is not as consistent for the other two terms as it appears there is a trade-off between them. In most instances, an average set of lower PP$^+$ values results in higher PP$^-$ values when comparing MoE models to their decoder-only counterparts.

From this, we would conclude that our proposed personalization scheme appears to consistently improve held-out log-likelihoods primarily due to better modeling of the sequential mark distributions, whereas it is a mixed bag for improving the distributions over event timings.

### B.3 Next Event Predictions

As described in Mei and Eisner [2017], we minimized the Bayes risk to determine decisions for what a predicted next time $\hat{t}_{i+1}$ and mark $\hat{k}_{i+1}$ would be after conditioning on a portion of a sequence $\mathcal{H}_{t_i} = [(t_1, k_1), \dots, (t_i, k_i)]$.

For the former prediction ($\hat{t}_{i+1}$), we first note that the next event time $t_{i+1}$ has a density $p_{i+1}(t) = P(t_{i+1} = t \,|\, \mathcal{H}_{t_i}) = \lambda(t|\mathcal{H}_t) \exp\left\{ -\int_{t_i}^{t} \lambda(s|\mathcal{H}_s)ds \right\}$. We define the precitec time $\hat{t}_{i+1}$ as the expected time under $p_{i+1}(t)$, i.e., $\hat{t}_{i+1} = \mathbb{E}[t_{i+1} \,|\, \mathcal{H}_{t_{i+1}}] = \int_{t_i}^{\infty} t p_i(t)dt$. With this definition, $\hat{t}_{i+1}$ minimizes the expected $L_2$ distance $\mathbb{E}_{p_{i+1}(t)}[(t - \hat{t}_{i+1})^2]$.

The latter prediction ($\hat{k}_{i+1}$) is computed similarly via $\hat{k}_{i+1} = \mathrm{argmax}_k \int_{t_i}^{\infty} \frac{\lambda_k(t|\mathcal{H}_t)}{\lambda(t|\mathcal{H}_t)} p_i(t)dt$. Note that this prediction of $k_{i+1}$ does not condition on the predicted time. These can be viewed as marginal predictions.

As mentioned previously, these integrals are approximated using an MC estimate with 10,000 samples.

**Time Prediction Results** Results for the above mentioned next event time prediction task can be seen in Table B. While the results seem slightly in favor of the proposed MoE models compared to their decoder-only counterparts, they are by no means conclusive. What is important to note is that the results are, for the most part, similar between the two types of models. This indicates at the very least that the addition of personalization for a given neural MTPP will not harm its predictive power for timings of events (whereas it would consistently improve prediction performance of marks as seen previously).

### B.4 Sampling Experiments

When modeling complex data with probabilistic models, having high log-likelihood scores does not always imply that the model will generate good samples [Theis et al., 2015]. We therefore describe experiments below that directly measure the sampling performance of the proposed models and baselines.

**Sampling Task** We sample future "trajectories" (sequences of event times and marks) for different models, conditioned on a partial history of a sequence (of relative size $\rho, 0\% \le \rho \le 100\%$), and evaluate the quality of the sampled trajectories relative to the observed actual future trajectory for the same sequence.

Table C: (a) Mean Jacard distances and (b) mean Wasserstein distances for samples generated on the four datasets for varying percentages of sequences to condition ($\rho$) across the four model variants. Lower values are better for both, and bolbed values indicate superior performance between an MoE model and its decoder-only counterpart.

| | | (a) Mean Sample Jacard Distances | | | | | | (b) Mean Sample Wasserstein Distances | | | |
|---|---|---|---|---|---|---|---|---|---|---|---|
| Dataset | $\rho$ | MoE-NHP | NHP | MoE-RMTPP | RMTPP | Dataset | $\rho$ | MoE-NHP | NHP | MoE-RMTPP | RMTPP |
| Meme | 10% | **0.410** | 0.455 | 0.483 | **0.447** | Meme | 10% | **26.302** | 27.133 | 28.296 | **28.022** |
| | 30% | **0.390** | 0.409 | 0.452 | **0.418** | | 30% | **21.477** | 22.862 | 23.768 | **23.484** |
| | 50% | **0.360** | 0.374 | 0.414 | **0.374** | | 50% | **17.657** | 18.204 | **18.547** | 18.842 |
| Reddit | 10% | **0.699** | 0.781 | **0.695** | 0.768 | Reddit | 10% | 22.735 | **22.145** | **34.095** | 34.408 |
| | 30% | **0.691** | 0.723 | **0.673** | 0.716 | | 30% | **18.765** | 18.826 | **25.019** | 25.113 |
| | 50% | **0.650** | 0.677 | **0.656** | 0.684 | | 50% | 16.137 | **15.683** | 18.513 | **18.065** |
| Amazon | 10% | **0.809** | 0.848 | **0.812** | 0.859 | Amazon | 10% | **4.523** | 4.852 | **4.546** | 4.733 |
| | 30% | **0.817** | 0.838 | **0.809** | 0.837 | | 30% | **3.718** | 4.020 | **3.979** | 4.046 |
| | 50% | **0.830** | 0.836 | **0.827** | 0.841 | | 50% | **3.298** | 3.392 | **3.525** | 3.636 |
| LastFM | 10% | **0.534** | 0.556 | **0.503** | 0.530 | LastFM | 10% | **3.755** | 3.822 | 3.402 | **3.351** |
| | 30% | **0.526** | 0.544 | **0.503** | 0.515 | | 30% | 3.132 | **3.015** | **2.793** | 2.843 |
| | 50% | **0.552** | 0.556 | **0.540** | 0.544 | | 50% | 2.443 | **2.426** | 2.639 | **2.568** |

More explicitly, for a given (real) test sequence $\mathcal{H}^s \in \mathcal{D}_{\text{Test}}$, let $\mathcal{H}^s_\pi$ be a portion of that sequence where $\pi < T$ is the smallest value that makes for $|\mathcal{H}^s_\pi| \approx \rho|\mathcal{H}^s|$ for $\rho \in [0,1]$. This partial, ground-truth sequence will be what we condition the model on and new events $(\hat{t}, \hat{k})$ will be sampled from time $\pi$ up until time $T$. We will denote $\mathcal{H}^s_{>\pi}$ as the portion of the real sequence not conditioned on, and $\hat{\mathcal{H}}^s_{>\pi}$ as the collection of all sampled events.

We use two different metrics to compare sampled data $\hat{\mathcal{H}}^s_{>\pi}$ to actual data $\mathcal{H}^s_{>\pi}$. These methods are introduced in the following two paragraphs.

**Sampled Marks Quality**  The first metric is a measurement of common marks between the two subsequences known as Jaccard Distance:

$$\text{JD}(\mathcal{H}, \hat{\mathcal{H}}) = 1 - \frac{|\{k \in \mathcal{H}\} \cap \{\hat{k} \in \hat{\mathcal{H}}\}|}{|\{k \in \mathcal{H}\} \cup \{\hat{k} \in \hat{\mathcal{H}}\}|}. \tag{E}$$

JD values close to 1 indicate that the sampled out-of-distribution marks do not match well with the observed sequence (or the user that generated the sequence). Likewise, values close to 0 indicate appropriate marks being sampled. This metric is particularly useful for datasets with a very large number of marks, with individual sequences only containing a fraction of them.

**Sampled Times Quality**  The second metric is a measurement of how similar the empirical distributions of sampled timestamps are to the true timestamps. Here we use the Earth-movers (Wasserstein) distance, defined for two sequences as:

$$\text{WD}(\mathcal{H}_{>\pi}, \hat{\mathcal{H}}_{>\pi}) = \inf_{\gamma \in \Gamma(\{t\}, \{\hat{t}\})} \int_{[\pi,T] \times [\pi,T]} |t - \hat{t}| d\gamma(t, \hat{t}), \tag{F}$$

where $\{t\}$ and $\{\hat{t}\}$ are the empirical distributions of times in $\mathcal{H}_{>\pi}$ and $\hat{\mathcal{H}}_{>\pi}$ respectively, and $\Gamma$ is a set of joint probability distributions whose marginals are $\{t\}$ and $\{t'\}$. WD values close to 0 indicate the two distributions are well-aligned both in the timing of events and the number of them. Larger WD values indicate that the sampled times are less congruent with the original times for the given sequence, or more broadly, for the given user.

**Sampling Results**  We evaluated the two metrics averaged over 1000 randomly selected test sequences from all four datasets for $\rho$ values of 10%, 30%, and 50%.

Table C shows the results. For JD, it appears that our personalization framework yields superior matching of the true mark mark distribution compared to the non-personalized, baseline models in 21 out of 24 settings. Similarly personalized models are only superior for mean WD values in 15 out of 24 settings. These findings further enforce our previous results for next event prediction, insofar as that the personalization framework appears to benefit mark distributions consistently versus yielding occasionally bettter modeling of the temporal dynamics.