[Reviews · NeurIPS 2020]

Review 1

Summary and Contributions: Post-rebuttal Comments: Thanks to the author for clearly responding to my questions. I suggest including aspects of their response (such as how to interpret latent variables) in the revised version of the paper. ------------------------------------------------------- This paper studies a model for event streams involving various types of events, for tasks such as model fitting and prediction. The novelty is a model that explicitly incorporates the aspect of the dataset where there is a set of event streams for different “users”, such as patients, systems, etc. The modeling work seems to be quite involved from an implementation perspective, relying on a latent variable for encoding user characteristics. The proposal employs a mixture-of-experts approach as well as variational autoencoders. Experiments are conducted on real-world datasets, demonstrating benefits of the proposed “user-dependent” approach. Overall, I think the paper brings new ideas to temporal point process models and considers a practically relevant problem. Although there is no new theory, the proposed ideas and methodology seem to be useful and show promising results.

Strengths: 1. The main novelty in the paper is around how to model different users and learn latent characteristics by just observing their event occurrences. I found the ideas reasonable but also somewhat complicated and sometimes hard to follow. The contribution relies heavily on the particular combination of ideas that the authors have connected, apparently effectively. 2. The experimental section seems reasonably convincing to me, showing the promise of latent variables for users.

Weaknesses: 1. The paper does not develop any new theory, which is not a major drawback in my view since there are sufficient methodological and experimental contributions. 2. I found parts of the paper difficult to follow but I think the paper is generally well written, so I have no specific suggestions (besides those in my detailed comments later). There are a lot of ideas to convey and not a lot of space, so I sympathize with the authors’ choices in this regard.

Correctness: I am not 100% sure but I do not see any reason to doubt correctness.

Clarity: Yes I think the paper is reasonably well written, but it's still occasionally hard to follow.

Relation to Prior Work: Please see my detailed comments.

Reproducibility: Yes

Additional Feedback: Additional Comments/Questions/Suggestions: P1, abstract: the tasks of event ranking and source identification are too vague in the abstract; they should be further clarified. P1: There is a lot of emphasis on neural point processes, which is understandable, but also on Hawkes processes, which is a bit less understandable. If the Hawkes parametric model is to be mentioned, I also suggest citing paper like: • Poisson-networks: A model for structured point processes, Rajaraman et al. 2005 • A model for temporal dependencies in event streams, Gunawardana et al. 2011 • Proximal graphical event models, Bhattacharjya et al, 2018. P1, line 25: “even” -> “event”. P2, line 73: the authors mention a hierarchy, but how is this a hierarchy? P3: In their setup, it looks like there are no features of the users available, or if they are, they are not exploited in the model. Could the authors confirm if that’s correct? P3: In equation 2, I did not understand why the notation concatenates z^u with the mark but keeps the timing separate. P3, para starting line 98: I found this confusing because the authors did not mention the conditional intensity function here, instead deferring it to later. I think they need to mention it earlier, somewhere around here. P4: What is the dimension of z^u? It appears to be a real-valued vector. Also, could the authors discuss the interpretation of this latent variable? P5, line 177: I did not understand this business about a single sample – how would it be possible to estimate this with a “single” sample? If the authors meant a single Monte Carlo computation, wouldn’t it be possible to do multiple samples but use a lower budget for each Monte Carlo, if there is a concern about computational time? I must be missing something here. P6, line 207: “define” -> “defined”. P7: Could the authors say more about why they think timing performance I the prediction performance is poor? P8: A longer conclusions section is warranted. P9 and onward: There are many inconsistencies around how references have been cited. This needs to be cleaned up.


Review 2

Summary and Contributions: Post-discussion update: Thanks to the authors for the thoughtful rebuttal. After reading all other reviews and the author response, I decide to keep my current score. Although I still think the weakness I pointed out is valid, I indeed appreciate the authors carefully put these techniques together and demonstrated the effectiveness of entire framework with extensive experiments. I think the paper should be accepted. ++++++ The paper proposes to learn user-dependent embeddings, on which the future event prediction is conditioned, by amortized variational inference. To generate the embedding for each user, the proposed framework considers all the ``reference sequences'' of that user by using a mixture of experts: each ``expert'' defines a proposal distribution over the embedding given one reference sequence.

Strengths: The proposed model is well-motivated and well-suited for the task of personalized modeling and prediction. Moreover, the application of VAE to continuous-time seq model has not been proposed in previous work yet. A reasonable amount of caution was paid into the model design, e.g., the mixture model to incorporate the user heterogeneity. Precisely, when ``reference sequences'' for a particular user are available, the proposed framework would define a proposal distribution over the user-level latent embedding for each reference sequence---which thus captures the heterogeneity of that user---and then average those proposal distributions to aggregate the heterogeneity for that user. The proposed method seems to be effective. It is augmented to two strong neural point processes (i.e., Du et al 2016 and Mei & Eisner 2017) and those augmented models achieve better performance (log-likelihood and prediction accuracies) on multiple datasets. Extensive experiments were done and there are more results in appendix.

Weaknesses: The contribution is incremental. Although it is a carefully thought framework, the use of VAE on neural sequential models in general has been widely proposed and thus there is no innovation on that side. As I mentioned in the [Strengths], the difference of this work from existing VAE+seqmodel work is that it applies VAE to continuous-time seq models. I have never seen such work yet and, for this reason, this work is good to show up in the community. However, we should admit that nothing has to be done particularly to incorporate ``continuous-time'' because that has been done by the neural models that this work augments to. The use of mixture of experts is an innovation on top of direct application of VAE but it is relatively minor compared to what can solidly warrant a NeurIPS paper.

Correctness: This is an empirical methodology and it doesn't have theoretical guarantees. So there is no clear notation of ``being correct''. Generally speaking, the proposed method is reasonable and the related math derivation is correct. The claims about its superiority has been supported by its experiments.

Clarity: The paper is clear in the sense that the key points are mostly clearly discussed with assistance of useful illustrative figures (e.g., figure-1 and 2). The paper is not clear enough in the sense of occasionally misusing statistical terms. E.g., ``sampled from the likelihood'' in line-99. You sample from a ``distribution'', not ``likelihood'', while ``likelihood'' is a function of model parameters and equal to the prob of data under the model. E.g., all terms in eqn-9 are cross-entropy so only using ``CE'' for the type term isn't very accurate.

Relation to Prior Work: It has clearly discussed relations with previous work.

Reproducibility: Yes

Additional Feedback:


Review 3

Summary and Contributions: This paper presents an approach to making personalized predictions for data that are generated by a collection of marked temporal point processes (MTPP) with different underlying structure. The learning problem enables conditioning on past data from other users, past data from the target user, and a temporal prefix from the target user to forecast the MPP into the future while reflecting inferred user-specific structure.

Strengths: + The problem proposed is relevant and timely. There are a number of important sources of data that match the assumed structure used in the paper including some forms e-commerce data as well as some forms of electronic medical record data. + The paper provides a modeling capability that has not been widely explored in this space. The approach is technically sound and the results are promising across a range of data sets.

Weaknesses: + The approach is a fairly straightforward combination of an MTPP model and a VAE via making the VAE latent state a feature in the MTPP model. There are other very similar constructions in other areas of supervised learning, although this specific contribution appears to be new. The novelty of the approach is thus moderate. + The framing of the source identification problem could benefit from refinement. In the mentioned applications (online shopping, social media, recommender systems) there is not an obvious practical issue with knowing which user generated which data. Some more discussion of why this problem is intrinsically useful, or perhaps why it illustrates interesting properties of the approach would be a helpful addition.

Correctness: * The empirical claims are consistent with the provided evidence. * As noted above, the method assembles several known modeling approaches and learning uses established methods, so there is no concern with correctness of the approach.

Clarity: The paper is clearly written. All of the figures are clear and easily readable.

Relation to Prior Work: The discussion of recent related work on MTPPs is clear. The discussion could be expanded slightly to include other time series modeling problems like time series clustering and time series factorization. For example, Bayesian Poisson Tensor Factorization deals with many of the same themes as this work, but with dyadic event data [1]. [1] https://arxiv.org/pdf/1506.03493.pdf

Reproducibility: Yes

Additional Feedback: See above for specific comments and suggestions. Post-Discussion Update: I thank the authors for their response. The question regarding the source identification problem has been addressed.


Review 4

Summary and Contributions: This paper proposes a personalized marked temporal point process model. In particular, in order to achieve the personalized prediction based on MTPP, this paper adopts the variational mixture-of-experts autoencoder to learn the user embeddings given the limited reference set. Then, the paper adopts the existing neural MTPP models, such as RMTPP and NHP, to predict the future time and markers by incorporating the user embeddings as inputs. The experimental results on four datasets show that the proposed models with user embeddings achieve better performance than models without using user embeddings.

Strengths: This paper targets on an interesting scenario, personalized neural temporal point processes. The proposed approach is clear and easy to follow.

Weaknesses: The proposed approach consists of two parts, training user embeddings and neural temporal point processes. For learning the user embeddings, this paper adopts the existing VAE-based approach (Shi et al, 2019), while for the neural temporal point processes, this paper also borrows the existing models. It is not clear to me how hard to combine these two parts together. For the experimental settings, it would be better to report the size of the reference sequences for each experiment. Besides tuning the training data sizes, it is also important to learn how the performance change via tuning the size of reference sequences. The two baselines are not strong enough. If the paper emphasizes the novelty in the MoE model, it would be better to compare with other approaches that can train the user embeddings or even uses randomly generated user embeddings.

Correctness: The proposed approach makes sense.

Clarity: Overall, the paper is well written and easy to follow. For Equation 7, it would be better to provide the explicit equation about how to derive the integral term, such as the equation given in the RMTPP paper.

Relation to Prior Work: Yes

Reproducibility: Yes

Additional Feedback:

[Author Response · NeurIPS 2020]

We thank the reviewers for their time and effort.

We begin by clarifying the scope and novelty of our contributions. Broadly, our work is a general extension of RNN-
based MTPP models (such as the RMTPP, but also the Neural Hawkes Process and others). More specifically: (1) our
work is the first to address the problem of personalizing neural MTPP models; (2) we combine VAE and neural MTPP
approaches in a non-trivial fashion (e.g., training with cyclical annealing); (3) we provide extensive experimental results
on multiple real-world datasets that show consistent and significant performance improvements, and (4) we provide a
detailed breakdown of where personalization helps in prediction (i.e., particularly at the beginning of sequences). (In
our paper we removed a list of contributions from the work to save space, but will add this back in the revised version).

We appreciate all of your comments and critiques about our work's clarity (as mentioned by Reviewers 1, 2, and 4),
citations and missing related works (Reviewers 1 and 3), and typos / terminology misuse (Reviewers 1 and 2). We will
be sure to incorporate this feedback into the camera-ready version. Listed below are our specific responses to all other
comments made. For brevity, many of them are paraphrased.

**Reviewer 1**   *How are the three different information sources (on line 73) a hierarchy?* We realized that our language
here is not precise. We were trying to say that the data is organized first as users, each of which have multiple event
sequences, with each sequence having a prefix and a future trajectory. We will refine these comments in the revised
version to avoid possible confusion.

*Are user features available?* Great question! For some of the datasets, user features were available to some degree
(e.g. a username, user-entered bio, etc.), but were not used for uniformity. In practice, user features would be a useful
addition to our approach and should be straightforward to add, e.g., by embedding this information and concatenating
with our user embeddings.

*In Eq. 2, why concatenate $z^u$ to the mark embedding and not the timing?* Another great question! We wanted our pro-
posed framework to be applicable regardless of base neural MTPP model. As such, all neural MTPPs represent marks
via embeddings which would allow us to concatenate the user embedding to it without problems. On the other hand,
different models incorporate the event timings differently where some embed it and others require using a scalar. As
such, there was no way to feasibly incorporate the user embedding to the timings in a universal manner.

*Dimensionality of $z^u$? Interpretation?* $z^u$ is a real-valued vector, and the dimensionality ranges from 32 to 64 depending
on the dataset (see supplement for more information). This vector can be interpreted as the sequence and user-specific
dynamics for a single history of events. $p(z^u)$ represent the various modes of dynamics for a given user $u$. For future
work it would be interesting to investigate using $z^u$ for downstream tasks, such as clustering users.

*Clarify why a "single sample" is used in the loss term?* We found using one (five) sample(s) for a Monte-Carlo estimate
of the expected value in the loss term to be sufficient for training (testing). We did this for computational efficiency as
each additional sample is tied to processing another event sequence.

*Why is the predicted timing performance poor?* We believe there may be a misunderstanding as the timing performance
for the personalized model is not poor, but rather just not that different to the baseline model performance (see more on
lines 255-259). As for why this is, we hypothesize that this could be because (i) the variation between users lies in the
subset of marks that occur for them rather than the timing or (ii) the temporal information in the encoding steps is not
being adequately captured which could be better enforced via regularization.

**Reviewer 2**   Please refer to our introductory statements as we believe this should hopefully address your concerns.

**Reviewer 3**   *Why is source identification meaningful / practical?* This is a practical problem in a number of appli-
cations, e.g., when trying to match clickstreams of non-logged-in users to known users, or for fraud detection in
cybersecurity. While our experiments are not meant to replicate a real-world application, we nonetheless believe the
experiments provide a useful way to evaluate and compare MTPP models.

**Reviewer 4**   *Report the reference sequence sizes?* The reference and target sequences have the same time window
and follow the same distribution of number of events (see "Mean $|\mathcal{H}|$" column in Table 1).

*Performance as a function of reference sequence size?* This is an interesting point. As of now, the model expects
reference sequences that span similar lengths of time that the target sequence does. This setup reflects how the
framework would typically be used in practice; however, restricting the reference information would be a way to test
generalization to longer sequences. Do note that we do analyze performance as a function of the *target* sequence size.

*"The two baselines are not strong enough"* We respectfully disagree. We believe that in conjunction with our strong
and extensive experimental findings that these two powerful baselines are sufficient at establishing a reference point to
observe how incorporating latent user embeddings improves modeling performance.

*"Better to compare to other methods that can train user embeddings or use randomly generated user embeddings."*
To the former point, expanding on Lines 184-185, we are especially interested in scenarios with brand new users.
That means that conventional means of having a set of explicitly learned user embeddings (as opposed to amortized
embeddings) would not be applicable as there would be no associated embeddings at test time. For the latter suggestion,
we are not quite sure how this implementation would be better from one without user embeddings at all as at best the
model would learn to ignore the randomized embeddings.

[Meta-Review · NeurIPS 2020]

Four knowledgeable referees support acceptance for the contributions, and I also recommend acceptance. However, please consider revising your paper to address reviewer's remarks and incorporate the changes you have promised in the rebuttal.